# HLA-B27 Status in Rheumatic Diseases: Clinical and Immunological Differences Between Positive and Negative Patients—A Comparative Study

**DOI:** 10.3390/biomedicines13081996

**Published:** 2025-08-16

**Authors:** Gabriela Isabela Răuță Verga, Nicoleta-Maricica Maftei, Andreea Eliza Zaharia, Carmen Loredana Petrea (Cliveți), Mariana Grădinaru Șerban, Diana-Andreea Ciortea, Alexia Anastasia Ștefania Balta, Ciprian Dinu, Doina Carina Voinescu

**Affiliations:** 1Faculty of Medicine and Pharmacy, University “Dunarea de Jos” of Galati, 800008 Galati, Romania; andreea.zaharia@ugal.ro (A.E.Z.); carmen.petrea@ugal.ro (C.L.P.); diana.ciortea@ugal.ro (D.-A.C.); alexia.balta@ugal.ro (A.A.Ș.B.); ciprian.dinu@ugal.ro (C.D.); doina.voinescu@ugal.ro (D.C.V.); 2Emergency Clinical Hospital for Children “Sf Ioan”, 800487 Galati, Romania; mariana_gradinaru@yahoo.com; 3Emergency Clinical Hospital for Children “Maria Sklodowska Curie”, 041451 Bucharest, Romania; 4St. “Apostol Andrei” County Emergency Clinical Hospital, 800578 Galati, Romania

**Keywords:** HLA-B27 status, rheumatic diseases, clinical differences, immunological markers, HLA-B*27-positive vs. -negative, inflammatory biomarkers

## Abstract

**Background/Objectives**: Human leukocyte antigen B27 (HLA-B27) is a genetic marker strongly associated with various inflammatory rheumatic diseases, particularly those within the spondyloarthritis spectrum. Its presence influences disease onset, clinical severity, and therapeutic strategies. However, comparative data between HLA-B*27-positive and -negative patients, especially in Eastern European populations, remain limited. The study aimed to investigate the clinical, paraclinical, and psychosocial differences between HLA-B*27-positive and -negative individuals diagnosed with rheumatic diseases, in order to better understand the implications of HLA-B27 status on disease expression and patient quality of life. **Methods**: A cross-sectional, observational study was conducted between June 2023 and December 2024 at the Emergency Clinical Hospital for Children “Sf Ioan” in Galati, Romania, in collaboration with “Dunarea de Jos” University. Fifty adult patients with various rheumatic conditions were enrolled and stratified into HLA-B*27-positive (n = 22) and -negative (n = 28) groups. Data collection included clinical evaluations, laboratory biomarkers (CRP = C-reactive protein; ESR = erythrocyte sedimentation rate), and a structured quality-of-life questionnaire. Statistical analysis was performed using SPSS v27. **Results**: HLA-B*27-positive patients were significantly younger (mean age 46.00 vs. 55.07 years, *p* = 0.018) and had higher CRP levels (>1 mg/dL in 53.33% vs. 0%, *p* = 0.001). Ankylosing spondylitis was more prevalent in this group (22.73% vs. 3.57%, *p* = 0.039). Magnetic resonance imaging (MRI) was more frequently used (68.18% vs. 39.29%, *p* = 0.042), indicating greater suspicion of axial involvement. HLA-B27-positive patients also reported higher perceived stress (mean score 2.41 vs. 1.21, *p* < 0.001). **Conclusions**: HLA-B*27 positivity is associated with earlier disease onset, increased systemic inflammation, greater axial involvement, and higher psychological stress. These findings emphasise the need for personalised, multidisciplinary care that integrates both medical and psychological support for HLA-B*27-positive patients.

## 1. Introduction

The human leukocyte antigen B27 (HLA-B27) is a class I surface antigen encoded by the major histocompatibility complex (MHC) on chromosome 6, known for its strong association with several inflammatory rheumatic diseases, particularly the group of seronegative spondyloarthropathies (SpA) [1]. The role of HLA-B27 as a genetic marker has been extensively investigated, yet important clinical and immunological differences persist between HLA-B27-positive and HLA-B27-negative individuals diagnosed with similar rheumatologic conditions [2].

Several large-scale cohort studies have established a robust association between HLA-B27 positivity and ankylosing spondylitis (AS), with prevalence rates exceeding 90% among AS patients in some populations [3]. On the other hand, HLA-B27-negative patients diagnosed with AS often exhibit atypical disease courses, with later onset, lower spinal involvement, and fewer extra-articular features [4].

From an immunopathological perspective, HLA-B27 positivity is linked to enhanced Th17 immune responses, increased production of interleukin-17 (IL-17) and tumour necrosis factor-alpha (TNF-α), and a propensity toward auto-inflammatory mechanisms, including misfolded protein responses in antigen-presenting cells [5]. These mechanisms may explain the more aggressive disease phenotype observed in HLA-B27-positive patients, particularly regarding axial involvement, enthesitis, and uveitis [6].

Conversely, HLA-B27-negative patients, although they may be diagnosed with similar diseases (e.g., reactive arthritis, psoriatic arthritis, or undifferentiated SpA), often demonstrate a more heterogeneous clinical picture, with milder axial disease, greater peripheral joint involvement, and sometimes overlapping features with rheumatoid arthritis (RA) or connective tissue diseases [7]. This heterogeneity has raised questions about whether HLA-B27-negative spondyloarthropathies represent distinct clinical entities rather than milder versions of the HLA-B27-associated forms.

Moreover, biomarker expression also varies between the two groups. HLA-B27-positive individuals tend to show higher levels of CRP and ESR during flares, reflecting more pronounced systemic inflammation [8]. On the other hand, HLA-B27-negative patients may have normal or only slightly elevated inflammatory markers, thus complicating the diagnosis and treatment monitoring [9].

Studies focusing on quality of life (QoL) and functional independence have also noted significant differences. HLA-B27-positive patients, particularly those with longstanding axial disease, often report greater physical limitation, chronic pain, and psychological burden, especially in the presence of structural spinal changes [10]. In contrast, HLA-B27-negative individuals may maintain higher levels of daily functioning, particularly when their disease remains peripheral or well-controlled through pharmacological management [11].

In terms of treatment response, some studies have shown that HLA-B27-positive patients may respond more favourably to anti-TNF therapy, possibly due to the immune-pathogenetic pathways involved [12]. However, HLA-B27-negative patients often require more individualised treatment approaches, as their response may vary depending on the underlying diagnosis and comorbidities [13].

Despite growing evidence, comparative studies remain relatively limited, especially in Eastern European populations, where genetic backgrounds and healthcare access can significantly influence both the expression and outcomes of rheumatic diseases.

### 1.1. Overview on the Epidemiology and Mechanisms of HLA B27

#### 1.1.1. Epidemiology of HLA-B27 in Rheumatic Diseases

The prevalence of the HLA-B27 allele varies significantly across different ethnic groups and geographic regions, reflecting both genetic background and population history [14]. Globally, HLA-B27 is present in approximately 6–8% of the general population, but this frequency ranges from less than 1% in some African populations to over 15% in Northern European and certain Indigenous Arctic communities [15].

The strongest epidemiological association is between HLA-B27 and ankylosing spondylitis (AS). In most populations, 80–95% of AS patients carry the HLA-B27 allele, compared to a much lower percentage in healthy controls [16]. This translates into a relative risk of developing AS that is approximately 50–100 times higher in HLA-B27-positive individuals compared to those without the allele [17].

However, HLA-B27 positivity is not synonymous with disease. Most individuals carrying the allele do not develop spondyloarthropathies, indicating that additional genetic, environmental, and immunological factors are required to trigger disease onset. The penetrance of the allele—i.e., the proportion of HLA-B27 carriers who develop AS—is estimated at 5–10%, depending on the population studied [18].

In diseases such as reactive arthritis, psoriatic arthritis, and inflammatory bowel disease-associated arthritis, HLA-B27 is present in 30–70% of patients, particularly in those with axial or enthesitis involvement [19]. Its presence in these conditions is associated with earlier disease onset, greater axial skeleton involvement, and higher recurrence rates following bacterial infections (e.g., *Chlamydia*, *Salmonella*) [20].

On the other hand, rheumatoid arthritis (RA) and systemic lupus erythematosus (SLE) show little or no association with HLA-B27 [21]. In fact, the HLA-B27 allele is often considered protective or neutral in these pathologies, which are more strongly linked to other MHC alleles, such as HLA-DR4 in RA and HLA-DR2/DR3 in SLE [22].

Romanian epidemiological data on HLA-B27 are relatively limited but align with broader European trends [23]. The allele is estimated to be present in approximately 6–9% of the Romanian general population, with higher rates reported among patients with ankylosing spondylitis and undifferentiated spondyloarthropathies [24]. There is growing clinical interest in early screening of HLA-B27 among individuals with inflammatory back pain or uveitis to facilitate early diagnosis and targeted treatment.

The clinical expression of HLA-B27 positivity also appears to be modulated by regional healthcare access, availability of biologic therapies, and patient awareness, all of which contribute to variability in disease burden and quality of life [25].

#### 1.1.2. Mechanisms of Action of HLA-B27

The role of HLA-B27 in the pathogenesis of spondyloarthropathies has been the subject of extensive investigation, yet its exact mechanisms of action remain incompletely elucidated. Comparative studies suggest that the immunological behaviour of this molecule significantly differs from that of other MHC class I alleles, contributing to both unique inflammatory cascades and specific clinical phenotypes in rheumatic diseases [26].

A prominent hypothesis centres around the tendency of HLA-B27 to misfold within the endoplasmic reticulum, leading to chronic activation of the unfolded protein response. While typical MHC molecules fold efficiently and exit the ER without incident, HLA-B27 has been shown to accumulate as misfolded intermediates, especially in the absence of β2-microglobulin [27]. This abnormal folding process has been implicated in enhanced intracellular stress, triggering IL-23 overexpression and expansion of pro-inflammatory Th17 cells [28]. Such effects appear to be more pronounced in HLA-B27-positive individuals with axial disease, particularly ankylosing spondylitis, where elevated IL-17 levels correlate with more aggressive disease progression [29].

By contrast, in HLA-B27-negative patients with similar clinical diagnoses, such cellular stress mechanisms are largely absent or minimal, which may partly explain the typically milder inflammatory profile, lower IL-17 expression, and reduced frequency of axial manifestations. These patients often show a distinct immunological pattern, more reliant on conventional cytokine responses such as TNF-α and IL-6 and sometimes overlapping features with seropositive diseases like rheumatoid arthritis [30].

Another key difference lies in the ability of HLA-B27 to form homodimers on the cell surface. These β2-microglobulin-free heavy chain dimers interact abnormally with natural killer (NK) cells and certain T-cell subsets via receptors such as KIR3DL2, potentially promoting sustained immune activation. This property is largely unique to HLA-B27 and has not been observed with other MHC class I alleles [31]. In HLA-B27-negative populations, the absence of these homodimeric forms suggests that alternative immunopathological mechanisms are at play, likely involving broader genetic influences and environmental modulators.

The theory of molecular mimicry also offers a comparative lens. HLA-B27 can present bacterial peptides that resemble self-antigens, which may lead to the activation of autoreactive cytotoxic T cells, particularly after infections with enteric or urogenital pathogens [32].

Furthermore, the cytokine environment differs markedly between the two groups. In HLA-B27-positive patients, upregulation of the IL-23/IL-17 axis plays a central role in enthesitis and new bone formation, key features of axial spondyloarthropathy [33]. Meanwhile, HLA-B27-negative patients tend to exhibit less enthesitis activity and more peripheral joint involvement, accompanied by lower IL-17 expression. This divergence in cytokine profiles reflects distinct underlying immunopathology, despite overlapping clinical labels [34].

In summary, comparative literature highlights fundamental differences in the molecular and immunological behaviour of HLA-B27-positive versus negative patients. Misfolding-induced ER stress, homodimer formation, antigenic mimicry, and preferential activation of the IL-23/IL-17 axis are mechanistic hallmarks of HLA-B27-associated disease, which are largely absent in HLA-B27-negative individuals. Understanding these distinctions is essential for both accurate disease classification and targeted therapeutic approaches.

Building upon the existing literature, it is clear that HLA-B27 functions not only as a genetic marker, but also as a modulator of immune response, disease expression, and prognosis in a variety of rheumatic conditions. The evidence consistently demonstrates that HLA-B27-positive individuals tend to follow distinct clinical, immunological, and functional trajectories compared to their HLA-B27-negative counterparts, particularly in the context of spondyloarthropathies and related inflammatory disorders [35].

Despite this growing body of knowledge, many comparative studies have focused narrowly on isolated diseases, such as ankylosing spondylitis or reactive arthritis, often overlooking the heterogeneous spectrum of rheumatic pathology encountered in everyday clinical practice [36]. Furthermore, real-world data integrating both objective inflammatory parameters and subjective quality-of-life outcomes remain limited, especially in Eastern European populations, where access to early diagnosis and biological therapies is variable.

In this context, the present study seeks to address these gaps by adopting a comparative, multidimensional approach, analysing the clinical, immunological, and psychosocial differences between patients with rheumatic diseases stratified by HLA-B27 antigen status. Through this lens, the study aims to clarify whether the presence of HLA-B27 is associated with distinct disease patterns, biomarker profiles, functional impairments, and differences in patient experience and health outcomes.

## 2. Materials and Methods

This research was designed as a cross-sectional observational study to capture an overview of the clinical, immunologic and quality of life characteristics of people with rheumatic diseases according to HLA-B27 antigen status. The study was carried out at the “Sf Ioan” Children’s Emergency Hospital in Galati, in collaboration with “Dunarea de Jos” University of Galati.

Data were collected between June 2023 and December 2024, during which patients who were presenting for routine rheumatology consultations or follow-up check-ups were invited to participate. The study targeted adult patients with different types of rheumatic disease, thus ensuring diversity in disease spectrum and severity. The main objective was to perform a comparative analysis between HLA-B*27 (genetic allele within the HLA system)-positive and HLA-B*27-negative patients, focusing on the following aspects:Levels of inflammatory markers;Degree of reported functional disability;Quality of life indicators including physical independence, impact of pain, and psychosocial burden.

Prior to inclusion in the study, all participants were fully informed about the purpose and procedures of the research and were asked to sign an informed consent form. Ethical approval was obtained from the Research Ethics Commission of the “Dunarea de Jos” University of Galati 270/CEU/15.05.2023.

All data has been anonymized and processed in accordance with the provisions of the General Data Protection Regulation (GDPR) to ensure complete confidentiality and protection of patients’ rights.

### 2.1. Participants

A total of 50 adult patients diagnosed with rheumatic diseases were enrolled in the study using a non-probabilistic, convenience sampling method. These patients were recruited during their routine rheumatologic evaluations or follow-up visits at the “Sf Ioan” Children’s Emergency Hospital in Galati, Romania. All patients included in the study underwent HLA-B27 antigen testing and completed a structured clinical and quality-of-life questionnaire.

Participants were divided into two comparison groups based on the results of the HLA-B27 antigen test:HLA-B*27-positive group: 22 patients (44%);HLA-B*27-negative group: 28 patients (56%).

This grouping allowed for a comparative analysis of clinical parameters, laboratory markers, and patient-reported outcomes between the two cohorts.

#### 2.1.1. Inclusion Criteria

Participants were eligible for inclusion if they met all the following criteria:Adults aged 18 years or older with a clinically confirmed diagnosis of a rheumatic disease (e.g., spondyloarthropathies, rheumatoid arthritis, osteoarthritis, lupus, etc.);Willingness to undergo HLA-B27 antigen testing;Completion and signing of a written informed consent form.

#### 2.1.2. Exclusion Criteria

The following exclusion criteria were applied to ensure data quality and ethical compliance:Refusal to undergo HLA-B27 testing;Participation in other concurrent clinical trials that could interfere with the data;Incomplete or missing informed consent documentation;Any condition that would impair the patient’s ability to understand or complete the study procedures (e.g., cognitive impairment, language barriers).

The final sample was composed of a heterogeneous group of rheumatologic patients representative of common autoimmune and inflammatory conditions encountered in clinical practice. All participants were anonymized and identified by numeric codes for data analysis.

### 2.2. HLA-B27 Detection

#### 2.2.1. Real-Time PCR Method

The HLA-B27 antigen status of each participant was determined by a molecular biology method based on the polymerase chain reaction (PCR) using the Bosphore^®^ HLA-B27 Detection Kit v2 (manufactured by Anatolia Geneworks, Istanbul, Turkey). This diagnostic test was chosen due to its high specificity and sensitivity in detecting the presence of the HLA-B*27 allele, which is strongly associated with certain autoimmune rheumatic diseases, especially spondyloarthritis.

A 3–5 mL peripheral venous blood sample was collected in EDTA tubes from each participant for HLA-B*27 detection by real-time PCR and DNA extraction were performed at the Clinical Analysis Laboratory of the “St. Ioan” Children’s Emergency Clinical Hospital, Galați.

DNA was extracted using STARMag 96 × 4 Universal Cartridge Kit. Seegene Inc., Irvine, California, USA. Amplification was performed by real-time PCR using a Bosphore^®^ HLA-B27 Detection Kit v2. Master Mix was prepared for all reactions (samples, positive and negative controls). The components of a resulting sample were as follows. A measure of 15 μL PCR Master Mix for each reaction, which was pipetted into each tube; a measure of 5 μL purified patients’ DNA or control template (positive/negative control) was added to reach a final reaction volume of 20 μL into each tube.

The Bosphore^®^ HLA-B27 Detection Kit v2 detects the existence of the HLA-B*27 allele in human biological samples and the allele is amplified; while fluorescence detection is accomplished using the FAM filter. Also, when the HLA-B*27 allele is absent in the sample, a reference gene (GAPDH) is amplified as an internal control. In order to ensure that the PCR is performed accurately, fluorescence detection of the GAPDH is performed using the Cy5 filter.

#### 2.2.2. Clinical Relevance and Limitations

While a positive HLA-B27 status suggests an increased genetic predisposition to certain autoimmune diseases (notably ankylosing spondylitis, reactive arthritis, and other spondyloarthropathies), it is not diagnostic on its own. Clinical correlation with symptomatology, physical examination, and imaging (e.g., MRI, radiographs) remains essential.

The Bosphore^®^ HLA-B27 Detection Kit v2 offers several advantages:High diagnostic accuracy;Rapid turnaround time;Minimal cross-reactivity with other HLA alleles.

However, its limitations include the following:Inability to differentiate among HLA-B27 subtypes;Risk of false negatives in cases of extremely low DNA concentrations or PCR inhibition.

Finally, interpretation must be contextualised within a full clinical rheumatologic workup.

All HLA-B27 testing was performed in the Molecular Biology Laboratory of Emergency Clinical Hospital for Children “Sf Ioan” Galati, under strict quality control and standard operating procedures.

### 2.3. Data Collection Instrument

A structured, interviewer-administered questionnaire was utilised as the primary tool for systematic data collection. The instrument was designed to gather both objective clinical information and subjective patient-reported outcomes relevant to the diagnosis, management, and daily impact of rheumatic diseases.

The questionnaire was developed in Romanian, the native language of the study population, to ensure clarity, accessibility, and full comprehension by all participants. Its content was validated internally through a multidisciplinary panel consisting of rheumatologists, immunologists, epidemiologists, and biostatisticians, who assessed its content validity, internal consistency, and usability in clinical settings.

The questionnaire was organised into three core domains.

1. Demographic and Clinical Characteristics

This section collected baseline sociodemographic and clinical data, including the following:Age;Sex;Area of residence (urban vs. rural);Level of education;Employment status;Smoking habits;Family history of rheumatic disease;Age at disease onset;Time elapsed between symptom onset and diagnosis;Current pharmacological treatments (NSAIDs, corticosteroids, DMARDs, biologics).

2. Patient-Reported Symptoms and Paraclinical Parameters

This section included the following:Duration and intensity of joint pain;Morning stiffness (duration and impact);Functional mobility limitations (e.g., walking, climbing stairs);Ocular, cutaneous, or gastrointestinal symptoms suggestive of extra-articular involvement.Paraclinical data from medical records were also collected, including the following:✓C-reactive protein (CRP);✓Rheumatoid factor (RF).

3. Health-Related Quality of Life (HRQoL) and Functional Independence

This domain assessed the subjective disease burden through the following:Self-reported pain levels (Visual Analogue Scale or Numeric Rating Scale);Perceived health status (poor–excellent);Impact of disease on daily functioning, work, and social activities;Degree of functional independence (using basic ADL indicators);Mental health and emotional well-being (anxiety, depression symptoms).

This multidimensional approach was intended to capture both the clinical profile and lived experience of patients with rheumatic diseases, facilitating a nuanced comparison between HLA-B*27-positive and -negative individuals.

All data were collected by trained clinical personnel during face-to-face interviews, ensuring the accuracy, consistency, and completeness of the responses.

### 2.4. Statistical Analysis

The data were processed and analysed using IBM SPSS Statistics, Version 27. Descriptive statistics such as means, medians, and standard deviations were applied to summarise both continuous and categorical variables, providing an overview of the sample characteristics and relevant clinical parameters.

To compare the two patient groups based on HLA-B*27 status, several inferential statistical tests were employed depending on the nature and distribution of the data. For continuous variables with normal distribution, the independent samples t-test was used, while the Mann–Whitney U test was applied for variables that did not meet normality assumptions. Categorical data were analysed using the chi-square test or Fisher’s exact test in cases with expected frequencies below the threshold for chi-square validity.

The strength of association between variables was further evaluated using Cramér’s V and the phi coefficient, particularly for binary or nominal-level data. The distribution of continuous variables was assessed through both the Kolmogorov–Smirnov and Shapiro–Wilk tests to ensure appropriate selection of parametric or non-parametric procedures.

Statistical significance was considered at a threshold of *p* < 0.05. In instances where *p*-values exceeded this threshold, but observable trends aligned with the study hypotheses, descriptive statistics and effect sizes were taken into account to support the interpretation of findings.

## 3. Results

### 3.1. Sociodemographic Profile of HLA-B27-Positive Versus -Negative Patients

To characterise the baseline demographics of the study population and assess possible differences between HLA-B*27-positive and -negative individuals, key variables such as age, sex, and environment of origin were analysed. The data revealed several noteworthy trends, some of which reached statistical significance, particularly in relation to age (Table 1).

Among the 50 patients included in the study, 44% (n = 22) were HLA-B*27-positive and 56% (n = 28) HLA-B27-negative. The overall mean age was 51.08 years (±13.61), with a median age of 50 years and an age range of 18 to 88 years. Notably, HLA-B*27-positive patients had a significantly lower mean age (46.00 ± 11.45 years) compared to their HLA-B*27-negative counterparts (55.07 ± 14.02 years), a difference that was statistically significant (t = 2.456, *p* = 0.018), (Figure 1).

The sex distribution showed a predominance of female patients, who accounted for 70% (n = 35) of the total cohort. This pattern was consistent across both HLA-B27 subgroups. Regarding environment of origin, 80% (n = 40) of patients lived in urban areas, with a higher percentage of urban residents in the HLA-B*27-positive group (90.91%) compared to the HLA-B*27-negative group (71.43%). Although not statistically tested, this difference may suggest variations in access to healthcare services or referral patterns between urban and rural populations.

### 3.2. Family History and Lifestyle Factors

In order to explore potential differences in lifestyle, anthropometric characteristics, and family history between patients with and without HLA-B*27 positivity, a comparative analysis was conducted. Table 2 presents a detailed summary of smoking habits, physical activity levels, body mass index (BMI) categories, and the presence of a family history of rheumatic disease in both subgroups. While most of the observed differences did not reach statistical significance, several trends emerged that may have clinical or epidemiological relevance, particularly in the context of disease awareness, health behaviours, and metabolic status.

Regarding smoking habits, 68% of all patients were non-smokers, 26% were active smokers, and 6% smoked occasionally. Among HLA-B*27-positive patients, 31.82% were smokers compared to 21.43% in the negative group. While this difference was not statistically significant, the higher smoking rate in the HLA-B*27-positive group could have implications for disease progression, especially in axial spondyloarthritis.

Only 28% of patients reported engaging in regular outdoor physical activity. Within the HLA-B*27-positive group, 36.36% reported such activity compared to 21.43% of HLA-B27-negative patients. Moreover, 45.46% of HLA-B*27-positive individuals engaged in exercise at least once per week, in contrast to only 21.42% among the HLA-B*27-negative group. This difference may indicate greater disease awareness or adherence to physiotherapy recommendations.

In terms of BMI, 50% of HLA-B*27-positive patients were of normal weight, compared to just 21.43% of those who were HLA-B*27-negative. Overweight status was identified in 27.27% of HLA-B*27-positive and 39.29% of HLA-B*27-negative individuals. Obesity (grade I or II) was more prevalent in the HLA-B*27-negative group (35.71%) than in the HLA-B*27-positive group (18.18%). These findings, though not statistically significant, suggest a more favourable metabolic profile among HLA-B*27-positive patients.

### 3.3. Clinical Symptoms and Rheumatologic Diagnoses

To further explore the clinical profile of patients according to HLA-B*27 status, a comparative analysis was conducted covering key aspects such as symptomatology at disease onset and at the time of evaluation, disease duration, joint involvement, rheumatologic diagnoses, and MRI referral rates. The following table summarises these findings, highlighting several patterns suggestive of a more inflammatory phenotype among HLA-B*27-positive individuals (Table 3).

Among HLA-B*27-positive patients, fatigue (72.73%), morning stiffness (54.55%), and inflammatory back pain (36.36%) were more frequently reported at disease onset than in the HLA-B*27-negative group. Fever and weight loss also appeared more often in the positive group. At the time of evaluation, 72.73% of HLA-B*27-positive patients continued to experience persistent pain, and 68.18% still reported fatigue. On the other hand, non-inflammatory back pain and prolonged morning stiffness were more commonly reported by HLA-B*27-negative individuals.

The average symptom duration was shorter in the HLA-B*27-positive group (6.03 years) than in the negative group (8.52 years), although this difference was not statistically significant. Both groups had a median disease duration of 6 years.

Regarding joint involvement, 68.18% of HLA-B*27-positive patients had 2–4 joints affected, while 13.64% had no joint involvement. Among HLA-B*27-negative patients, 50% had 2–4 joints affected, and 28.57% reported no joint involvement.

Rheumatologic diagnoses differed between the groups. Ankylosing spondylitis was significantly more common in HLA-B*27-positive patients (22.73%) than in those who were negative (3.57%), with a statistically significant association confirmed by a chi-square test (χ^2^ = 4.281, *p* = 0.039) and a weak but meaningful phi coefficient (φ = 0.293). Other differences were noted for gonarthrosis (25.00% in negatives vs. 4.55% in positives) and coxarthrosis (25.00% vs. 9.09%).

Finally, MRI was more frequently used in the diagnostic process of HLA-B*27-positive patients (68.18%) compared to HLA-B*27-negative patients (39.29%), with a significant association (χ^2^ = 4.121, *p* = 0.042, φ = 0.287), suggesting greater suspicion of axial involvement in the HLA-B*27-positive subgroup.

### 3.4. Paraclinical Results: Inflammatory Markers and Imaging

To evaluate the inflammatory profile associated with HLA-B*27 status, CRP levels were assessed at both initial and latest clinical evaluations. Additionally, the frequency of MRI use and its diagnostic relevance were analysed. These parameters serve as surrogate markers for disease activity and clinical suspicion of axial involvement, particularly in spondyloarthropathies (Table 4).

At initial assessment, 40.00% of HLA-B*27-positive patients had CRP levels above 1 mg/dL, compared to only 9.09% in the HLA-B*27-negative group. On the other hand, moderately elevated CRP levels (0.5–1 mg/dL) were more frequent in HLA-B*27-negative patients (36.36%) than in the positive group (6.67%). The percentage of patients with normal CRP levels was similar in both groups, around 53–55%.

At the latest clinical evaluation, CRP elevation became more pronounced in the HLA-B*27-positive group, where 53.33% had values above 1 mg/dL. None of the HLA-B*27-negative patients had CRP above this threshold; instead, 66.67% had normal values and 33.33% had moderately elevated levels. This difference was statistically significant (χ^2^ = 13.529, *p* = 0.001), with a strong association (φ = 0.672), indicating a meaningful inflammatory difference between the groups (Figure 2).

MRI was performed in 68.18% of HLA-B*27-positive patients versus 39.29% of HLA-B*27-negative patients. This difference was statistically significant (χ^2^ = 4.121, *p* = 0.042) and the association was of weak-to-moderate strength (Cramer’s V = 0.287), suggesting greater clinical suspicion of axial involvement among HLA-B*27-positive individuals (Figure 3).

### 3.5. Treatment

In terms of treatment, 36.36% of HLA-positive patients are receiving a complex treatment with several groups of drugs and 45.45% are not receiving any treatment. In terms of treatment, 25% of HLA-negative patients receive a complex treatment with several groups of drugs and 32.14% do not receive any treatment. Patients were treated with nonsteroidal anti-inflammatory drugs (NSAIDs): 40.91% of HLA-positive patients (n = 9) and 50% of HLA-negative patients (n = 14); corticosteroids: 31.82% of HLA-positive patients (n = 7) and 21.43% of HLA-negative patients (n = 6), respectively; biologic treatment requiring hospitalisation: 22.73% of HLA-positive patients (n = 5) and 7.14% of HLA-negative patients (n = 2), respectively; and biologic treatment requiring hospitalisation: 4.55% of HLA-positive patients (n = 1) and 7.14% of HLA-negative patients (n = 2).

### 3.6. Quality of Life and Perceived Stress

To assess psychosocial burden and quality of life in relation to HLA-B*27 status, patients were asked about the impact of pain on their daily activities, sleep disturbances, and psychological stress associated with their antigen status. While pain and sleep disruption were commonly reported in both groups, only perceived stress showed a statistically significant difference, highlighting the psychological impact of being HLA-B*27-positive (Table 5).

Pain was a frequent symptom affecting daily functioning in both groups. A moderate impact was reported by 54.55% of HLA-B*27-positive patients and 46.43% of those who were HLA-B*27-negative. Slight impact was noted in approximately 32% of positive and 36% of negative patients, while major pain-related limitation was reported by 13.64% of HLA-B*27-positive and 17.86% of HLA-B*27-negative individuals. These differences were not statistically significant but indicate a slightly higher functional burden in the HLA-B*27-positive group.

Sleep disturbances were more common among HLA-B*27-positive patients, with 31.82% reporting frequent or very frequent sleep issues, compared to only 14.29% of HLA-B*27-negative patients. Conversely, 85.71% of HLA-B27-negative individuals reported stable sleep (occasional or no disturbance), compared to 68.18% in the HLA-B*27-positive group. Despite this trend, the difference did not reach statistical significance (*p* = 0.178).

The most significant difference emerged in the domain of perceived stress. HLA-B27-positive patients reported significantly higher stress levels, with a mean score of 2.41 (±1.297), compared to 1.21 (±0.686) among HLA-B*27-negative individuals. This difference was statistically significant (U = 142.50; Z = −3.843; *p* < 0.001), suggesting a meaningful psychological impact related to awareness of antigen status, disease prognosis, or symptom burden (Figure 4).

## 4. Discussion

The results obtained in this study contribute valuable insights to clinical understanding of the differences between HLA-B*27-positive and -negative patients; however, they should be critically assessed in light of the study design and methodological limitations. Although some findings reached statistical significance, it is important to acknowledge that the sample size was relatively small, which may affect the statistical power and the generalisability of the conclusions.

For instance, the observed differences in CRP levels and the frequency of MRI use among HLA-B*27-positive patients may partly reflect a clinician’s tendency to investigate more thoroughly those patients known to have a genetic predisposition to inflammatory diseases. As such, some results may be influenced by selection bias or unequal access to advanced imaging investigations. Moreover, the increased levels of stress reported by HLA-B*27-positive patients may not only be a consequence of symptomatology but also reflect a greater awareness of their prognosis, introducing a psychological element that is difficult to quantify objectively.

It is also worth noting that lifestyle-related variables, such as physical activity or BMI, were self-reported, which may introduce reporting or recall bias. The absence of significant differences in some of these areas does not rule out the presence of clinically relevant effects, which might become apparent in a larger sample or a longitudinal study.

While the study provides findings consistent with the existing literature, the interpretation of results should be approached with caution, taking into account the methodological limitations, potential sources of bias, and the need for validation within a broader clinical context.

The present study identified significant differences between HLA-B*27-positive and -negative patients, both clinically and paraclinically. These findings are consistent with the existing literature, which highlights the role of the HLA-B27 antigen in the pathogenesis and clinical expression of inflammatory rheumatic diseases [37,38,39].

HLA-B*27-positive patients exhibited an earlier onset of symptoms compared to their HLA-B*27-negative counterparts. This observation is supported by previous studies indicating an association between HLA-B*27 positivity and earlier onset of axial spondyloarthritis [40,41]. Furthermore, a higher prevalence of ankylosing spondylitis was observed among HLA-B27-positive patients, aligning with existing data that demonstrate a strong correlation between this antigen and ankylosing spondylitis [42].

From a paraclinical perspective, CRP levels were significantly elevated in HLA-B*27-positive patients. This inflammatory marker is commonly used in assessing disease activity in spondyloarthritis and other rheumatic conditions. Prior research has shown that HLA-B*27-positive patients tend to exhibit higher CRP levels, suggesting a more pronounced inflammatory response [43,44].

Magnetic resonance imaging (MRI) was more frequently employed in the evaluation of HLA-B*27-positive patients, possibly reflecting increased clinical suspicion of axial involvement in this group. The literature supports the use of MRI in the early diagnosis of axial spondyloarthritis, particularly in HLA-B*27-positive individuals, due to the modality’s enhanced sensitivity in detecting sacroiliac inflammation [45,46].

In terms of quality of life, HLA-B*27-positive patients reported higher levels of perceived stress. This may be attributed to greater awareness of disease prognosis or to more severe symptomatology. Previous studies have identified a higher prevalence of mood disorders, such as anxiety and depression, among patients with axial spondyloarthritis—particularly those who are HLA-B*27-positive [47].

Differences in lifestyle factors, such as physical activity levels and body mass index (BMI), were observed between the two groups, although these did not reach statistical significance. HLA-B27-positive patients reported higher levels of physical activity, which may reflect increased awareness of the role of exercise in symptom management. Additionally, a tendency towards lower BMI was noted in HLA-B27-positive patients, which may have implications for disease severity and treatment response.

In conclusion, the findings of this study are consistent with the specialist literature and underscore the significance of the HLA-B27 antigen in the clinical and paraclinical manifestation of inflammatory rheumatic diseases. Identifying these differences can enhance our understanding of disease pathogenesis and support the development of personalised therapeutic strategies for patients with axial spondyloarthritis.

The findings of this study provide additional evidence supporting the role of the HLA-B27 antigen not only as a genetic marker associated with specific rheumatic conditions, but also as a clinical indicator of a more active inflammatory phenotype. The observed differences in age of onset, elevated levels of inflammatory markers such as C-reactive protein, and the higher incidence of ankylosing spondylitis in the HLA-B27-positive group underscore the importance of this biomarker in patient stratification and the personalisation of therapeutic management.

From a biological perspective, HLA-B*27 positivity is associated with abnormalities in antigen processing, defective protein folding, and the activation of T cell-mediated inflammatory pathways, which may explain the higher levels of systemic inflammation observed in positive patients. This immunological hyperactivity manifests clinically as more intense symptomatology and potentially greater functional impairment—findings supported by the results of this study.

Clinically, the identification of HLA-B*27-positive patients allows for earlier diagnosis of conditions within the spondyloarthritis spectrum and may justify more rapid initiation of biologic therapies or closer monitoring of disease activity. Furthermore, the observation of increased psychological stress among these patients highlights the need for a multidisciplinary approach that integrates psychological support into the rheumatological treatment plan.

Overall, the clinical and biological relevance of the HLA-B27 antigen is reinforced by the present study’s data, which may contribute to improved strategies for diagnosis, prognosis, and therapeutic intervention in the field of rheumatology.

Although the present study offers valuable insights into the clinical and biological differences between HLA-B*27-positive and -negative patients, several limitations must be acknowledged that may affect the interpretation and applicability of the results. Firstly, the relatively small sample size limits the statistical power of the analyses and the generalisability of the findings to the wider population of patients with rheumatic diseases. A larger sample would not only allow for a more robust confirmation of the identified associations but also enable the exploration of additional variables, such as disease activity using validated scores or the impact of biologic therapies.

Secondly, the cross-sectional nature of the study restricts the ability to draw causal inferences. The observed relationships between HLA-B*27 status, clinical parameters, and inflammatory markers represent associations at a single point in time, without the capacity to evaluate disease progression or treatment response over time. Longitudinal studies are therefore needed to examine the differing clinical trajectories based on HLA-B*27 status.

Another important limitation arises from the fact that much of the collected data is self-reported, introducing potential recall or subjective interpretation biases—particularly concerning lifestyle factors, physical activity levels, perceived stress, and sleep quality. Incorporating objective assessment tools, such as actigraphy, validated disease activity scores, and standardised quality of life questionnaires, could enhance the accuracy and comparability of the data.

Looking ahead, the development of multicentric studies with larger samples and standardised methodologies is essential. These should encompass clinical, immunological, genetic, and imaging parameters. Of particular interest would be the exploration of psychoneuroimmunological mechanisms through which HLA-B*27 status influences not only symptomatology but also patients’ perception of their illness, associated stress, and quality of life. In addition, comparative evaluation of treatment responses to conventional and biological therapies, stratified by HLA-B*27 status, could yield valuable data to support personalised medicine in rheumatology.

The observational, cross-sectional design chosen for this study was appropriate for the proposed research objectives, allowing for the comparison of clinical, paraclinical, and psychosocial characteristics between patients with and without the HLA-B27 antigen. This type of design is commonly used in descriptive clinical research, particularly when aiming to explore associations between genetic biomarkers and clinical phenotypes in the absence of therapeutic intervention.

The rationale for selecting a cross-sectional model stemmed from the need to obtain an overview of the distribution of variables across two distinct groups within a real-world clinical setting. In the context of rheumatic diseases, where early diagnosis and risk profiling are priority goals, such a design enables the rapid collection of clinically and epidemiologically relevant data.

Another argument supporting the chosen model is its practical feasibility. Recruiting patients from a single centre over a defined period facilitated standardisation of evaluations and reduced inter-observer variability. Furthermore, the use of self-reported data was appropriate for investigating subjective dimensions such as quality of life, pain perception, and stress levels—factors that cannot be assessed through standard paraclinical methods.

An inherent limitation of this model lies in its inability to establish causal relationships between variables. However, the aim of this research was not to demonstrate causality but rather to identify clinical and biological patterns associated with HLA-B27 status. In this regard, the chosen design was methodologically appropriate and justified.

Therefore, the use of an observational, non-interventional model enabled a comprehensive and balanced exploration of the phenomenon under investigation, providing a foundation for future research hypotheses and the development of larger-scale prospective or experimental studies.

## 5. Conclusions

This study explored the clinical, demographic, and psychosocial differences between HLA-B*27-positive and -negative patients within a cohort of individuals diagnosed with various rheumatologic conditions. Although some findings did not reach statistical significance, several important trends emerged, particularly regarding inflammatory markers, clinical presentation, and quality of life.

HLA-B*27-positive patients were significantly younger than their HLA-B*27-negative counterparts (mean age 46.00 vs. 55.07 years; *p* = 0.018), supporting the known association between HLA-B*27 and earlier disease onset. Urban residency was more frequent in HLA-B*27-positive individuals (90.91% vs. 71.43%), potentially reflecting differences in access to diagnostic facilities and healthcare services. The distribution of sex and family history of rheumatic diseases was similar across groups.

Clinically, HLA-B*27-positive patients presented a more inflammatory profile. At disease onset, they reported higher rates of fatigue (72.73%), morning stiffness (54.55%), and inflammatory back pain (36.36%). These trends persisted at the time of assessment, with persistent pain and fatigue remaining prevalent. CRP levels at the latest evaluation were significantly higher in the HLA-B*27-positive group, with 53.33% exceeding 1 mg/dL compared to none in the negative group (*p* = 0.001; φ = 0.672). MRI use was also significantly more common in HLA-B*27-positive patients (68.18% vs. 39.29%; *p* = 0.042), reflecting a higher suspicion of axial involvement.

Ankylosing spondylitis was notably more frequent among HLA-B*27-positive patients (22.73% vs. 3.57%; *p* = 0.039), consistent with established immunogenetic patterns. Despite comparable disease duration and joint involvement, the HLA-B*27-positive group experienced a higher burden of sleep disturbances (31.82% vs. 14.29%) and significantly higher perceived psychological stress (mean score 2.41 vs. 1.21, *p* < 0.001).

HLA-B*27-negative rheumatic patients are often diagnosed late, which delays the initiation of appropriate treatment. These individuals may present with atypical clinical forms, with a more severe course and a variable response to therapy, emphasising the need for detailed clinical evaluation and more rigorous diagnostic criteria. The implementation of strategies for early diagnosis, focusing on early recognition of clinical phenotypes and continuing education of rheumatology specialists, is essential to improve the prognosis of these patients.

Overall, HLA-B*27 positivity was associated with earlier disease onset, increased inflammatory activity, greater diagnostic imaging use, and a higher psychological impact. These findings underline the need for integrated clinical and psychosocial management strategies in HLA-B*27-positive patients, particularly in axial spondyloarthritis.

## Figures and Tables

**Figure 1 biomedicines-13-01996-f001:**
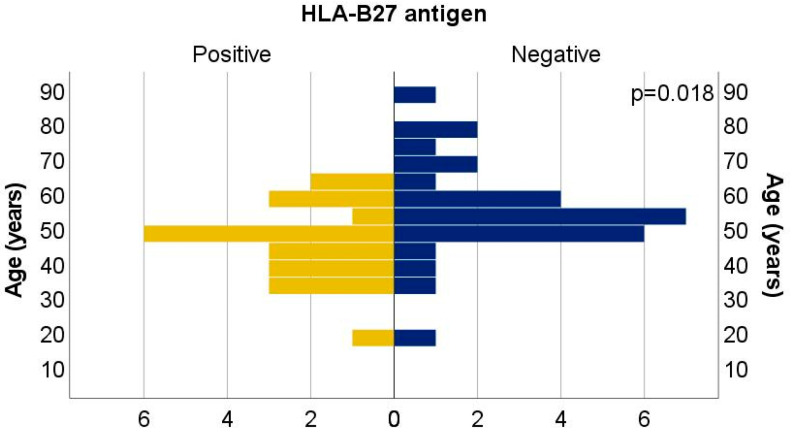
Distribution of patients by age and HLA-B27 antigen.

**Figure 2 biomedicines-13-01996-f002:**
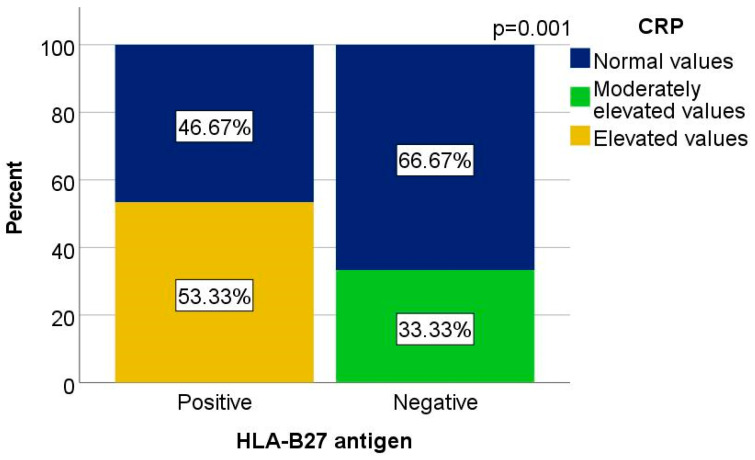
Distribution of cases according to current C-reactive protein level for patients who filled in the parameter value (at the time of filling in the questionnaire)—30 patients.

**Figure 3 biomedicines-13-01996-f003:**
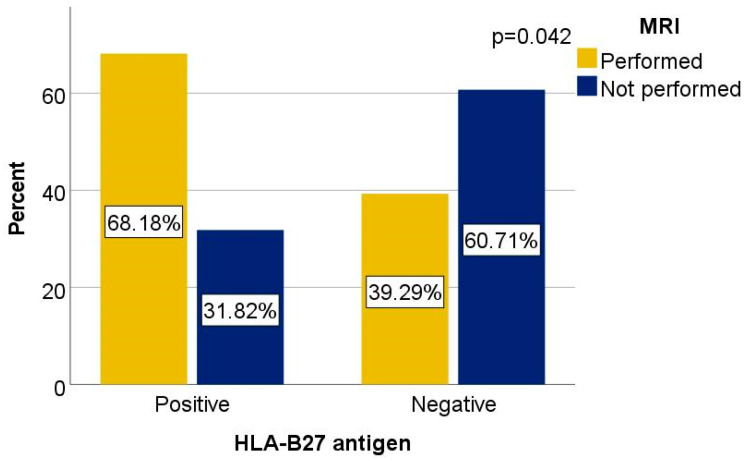
Distribution of MRI in HLA-B*27-positive and -negative patient groups.

**Figure 4 biomedicines-13-01996-f004:**
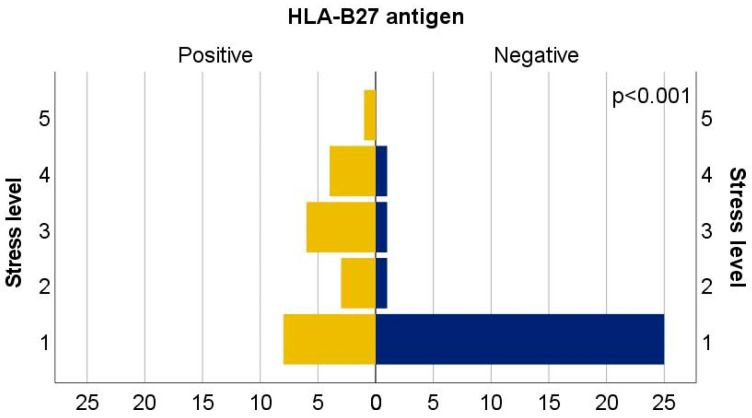
Distribution of cases by stress level between the HLA-positive and HLA-negative groups.

**Table 1 biomedicines-13-01996-t001:** Demographic and clinical characteristics of the study population by HLA-B27 status.

Characteristic	All Patients (n = 50)	HLA-B*27-Positive (n = 22)	HLA-B*27-Negative (n = 28)
HLA-B27 status	44% positive/56% negative	–	–
Age range (years)	18–88 years	18–64 years	20–88 years
Mean age ± SD (years)	51.08 ± 13.61	46.00 ± 11.45	55.07 ± 14.02
Median age (years)	50	48	53
Sex: Female	35 (70%)	16 (72.73%)	19 (67.86%)
Sex: Male	15 (30%)	6 (27.27%)	9 (32.14%)
Environment: Urban	40 (80%)	20 (90.91%)	20 (71.43%)
Environment: Rural	10 (20%)	2 (9.09%)	8 (28.57%)

**Table 2 biomedicines-13-01996-t002:** Lifestyle, family history, and anthropometric characteristics by HLA-B27 status.

Category	Variable	All Patients (n = 50)	HLA-B*27-Positive (n = 22)	HLA-B*27-Negative (n = 28)
Smoking Habits	Non-smoker	68%	14 (63.64%)	20 (71.43%)
	Smoker	26%	7 (31.82%)	6 (21.43%)
	Occasional smoker	6%	1 (4.55%)	2 (7.14%)
Physical Activity	Outdoor activity (yes)	28%	8 (36.36%)	6 (21.43%)
	Exercise ≥ 1/week	22%	10 (45.46%)	6 (21.42%)
	Exercise < 1/week or none	68%	12 (54.54%)	22 (78.58%)
BMI Category	Underweight	4%	4.55%	3.57%
	Normal weight	34%	50.00%	21.43%
	Overweight	34%	27.27%	39.29%
	Obesity Grade I	24%	18.18%	28.57%
	Obesity Grade II	4%	0.00%	7.14%
Family History of Rheumatic Disease	Positive	56%	10 (45.45%)	18 (64.29%)
	Negative	44%	12 (54.55%)	10 (35.71%)

**Table 3 biomedicines-13-01996-t003:** Clinical symptoms, disease characteristics, and diagnoses by HLA-B27 status.

Category	Variable	HLA-B*27-Positive (n = 22)	HLA-B*27-Negative (n = 28)
Initial Symptoms	Fatigue	72.73%	53.57%
	Morning stiffness	54.55%	35.71%
	Inflammatory back pain	36.36%	–
	Fever	9.09%	0%
	Weight loss	22.73%	7.14%
Current Symptoms	Persistent pain	72.73%	50%
	Fatigue	68.18%	46.43%
	Inflammatory back pain	31.82%	21.43%
	Non-inflammatory back pain	22.27%	32.14%
	Morning stiffness (long-lasting)	36.36%	35.71%
Symptom Duration	Mean ± SD (years)	6.03 ± 5.35	8.52 ± 8.10
	Median duration (years)	6	6
Number of Joints Affected	None	13.64%	28.57%
	1 joint	0	0
	2–4 joints	68.18%	50.00%
	>5 joints	9.09%	0%
Rheumatologic Diagnoses	Ankylosing spondylitis	22.73%	3.57%
	Rheumatoid arthritis	22.73%	17.86%
	Gonarthrosis	4.55%	25.00%
	Coxarthrosis	9.09%	25.00%
	Osteoporosis	22.73%	17.86%
	Psoriatic arthritis	0.00%	3.57%
	Lupus erythematosus	9.09%	3.57%
	Gout	4.55%	7.14%
	Vasculitis	4.55%	0.00%
	Undiagnosed	27.27%	32.14%
MRI Referral	Performed MRI	68.18%	39.29%

**Table 4 biomedicines-13-01996-t004:** Inflammatory markers, MRI use, and statistically significant associations by HLA-B27 status.

Parameter	Category	HLA-B*27-Positive	HLA-B*27-Negative	Statistical Significance
CRP: Initial Evaluation (n = 26)	Normal (<0.5 mg/dL)	53.33%	54.55%	*p* > 0.05
	Moderately elevated (0.5–1 mg/dL)	6.67%	36.36%	*p* > 0.05
	Elevated (>1 mg/dL)	40.00%	9.09%	*p* > 0.05
CRP: Latest Evaluation (n = 30)	Normal (<0.5 mg/dL)	46.67%	66.67%	χ^2^ = 13.529; *p* = 0.001
	Moderately elevated (0.5–1 mg/dL)	0.00%	33.33%	φ = 0.672 (strong association)
	Elevated (>1 mg/dL)	53.33%	0.00%	
MRI Referral	MRI performed	68.18%	39.29%	χ^2^ = 4.121; *p* = 0.042
				Cramer’s V = 0.287 (weak–moderate)
Summary of Associations	CRP > 1 mg/dL (latest evaluation)	53.33%	0.00%	Significant
	MRI performed	68.18%	39.29%	Significant

**Table 5 biomedicines-13-01996-t005:** Quality of life, sleep disturbance, and perceived stress by HLA-B27 status.

Category	Variable	HLA-B*27-Positive (n = 22)	HLA-B*27-Negative (n = 28)	Statistical Significance
Impact of Pain on Activities	Slight impact	~31.82%	~35.71%	*p* > 0.05
	Moderate impact	54.55%	46.43%	
	Major impact	13.64%	17.86%	
Sleep Disturbance	Frequent/very frequent sleep issues	31.82%	14.29%	*p* = 0.178 (not significant)
	Stable sleep (occasional/no disturbance)	68.18%	85.71%	
Perceived Stress (1–5 scale)	Mean stress score	2.41	1.21	*p* < 0.001 (significant)
	Median	2.50	1.00	U = 142.500; Z = −3.843
	Standard deviation	1.297	0.686	
	Sample size (n)	22	28	
Summary of QoL Findings	Pain impact	Slightly higher in positives		*p* > 0.05
	Sleep disturbance	More frequent in positives		*p* > 0.05
	Perceived stress	Significantly higher		*p* < 0.001

## Data Availability

Personal medical data are publicly unavailable due to privacy and ethical restrictions.

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
