# Peer review of "HLA-B27 Status in Rheumatic Diseases: Clinical and Immunological Differences Between Positive and Negative Patients—A Comparative Study"

_biomedicines, 2025, doi:10.3390/biomedicines13081996_

Round 1

Reviewer 1 Report

Comments and Suggestions for Authors

Abstract. As you first explain HLA abbreviation at the beginning of the abstract, you also should explain other abbreviations like CRP, ESR and MRI. It may help the understanding of readers not familiarized with that terms.

My first comment is about HLA nomenclature. HLA alleles (or groups of alleles) should be named like with an asterisk between the letter and the number. In example: HLA-B*27. Serologic groups of alleles are named without the asterisk and may include many different alleles within (i.e.: HLA-B8 splits first in HLA-B12 serotype which also splits in HLA-B44 and HLA-B45, including HLA-B*44 and HLA-B*45 alleles). Maybe, HLA-B27 nomenclature is accurate in the Introduction sections, where authors talk about the antigen in a more general way, but I suggest authors to change HLA-B27 nomenclature for HLA-B*27 for talking about HLA-B*27 detected allele in participants.

Introduction. Line 75, “tush” word may be a mistyping. It should be “thus”; Line 81, it should be “In contrast” at the beginning; Line 86, change “hawever” for “however”.

Methods section. Lines 282-291 (specially line 286), bullets are not well structured since authors start listing advantages of Bosphore® HLA-B27 Detection Kit v2m and continue listing limitations withouth any separation in the text as a different sublist; Lines 266-268, rewrite this phrase for better understanding; Lines 270 and 272, in “HLAB27” the hyphen is missing;

Results section. Table 1, change “pozitive” for “positive”; Lines 376 and 390, again the hyphen is missing in “HLAB27”; Lines 470, 479 and 524, change “figurea” for “Figure”; Line 481, again the hyphen is missing in HLA-B27.

Results section, lines 384-392. Marital status as well as studies level of participants are completely irrelevant for the development and the conclusions of this study. Delete this paragraph of results. Parameters like age, smokers, sex or obesity among others are directly related to authors’ results, but marital status or educational level are completely antithetical to these.

Results section, paragraph lines 483-497. I have to remind authors that alternative therapies lack scientific validation and may supose serious risks to patients, especially when used in place of proven medical treatments. Despite the fact that these methods have no clinical evidence supporting their safety or effectiveness, I think that including this paragraph in the results section is not relevant for the paper nor the conclusions.

Discussion and Conclusions sections are well structured and are concordant with Results section. No relevant changes have to be performed in these sections.

Reviewer 2 Report

Comments and Suggestions for Authors

The manuscript " HLA-B27 Status in Rheumatic Diseases: Clinical and Immunological Differences Between Positive and Negative Patients- A Comparative Study" is novel and logically written. This finding is meaningful in emphasizing the need for personalized, multidisciplinary care that integrates both medical and psychological support for HLA-B27 positive patients. It is good to be published whenever some grammar problems are addressed .

  1. There are some typos in the text, e.g., line 86, the "Hawever" should be "However".
  2. Please check the capitalization rules in titles, e.g., in is not capitalized in titles, etc. 

Round 2

Reviewer 1 Report

Comments and Suggestions for Authors

I thank the authors for correctly addressing all my comments. They have improved the paper and it is now suitable for publication in Biomedicines journal. I have only two little comments more about HLA-B27 nomenclature/correct writing:

Abstract conclusions and Keywords: It should be HLA-B*27 in both places, because authors are talking about HLA-B*27 allele positivity.

Material and Methods, Line 272. It should also be HLA-B*27 since you are referring to the allele.
